# Integrative Multi-Omics Approaches for Identifying and Characterizing Biological Elements in Crop Traits: Current Progress and Future Prospects

**DOI:** 10.3390/ijms26041466

**Published:** 2025-02-10

**Authors:** Bing-Liang Fan, Lin-Hua Chen, Ling-Ling Chen, Hao Guo

**Affiliations:** State Key Laboratory for Conservation and Utilization of Subtropical Agro-Bioresources, College of Life Science and Technology, Guangxi University, Nanning 530004, China; blfan@st.gxu.edu.cn (B.-L.F.); linhuachenchn@outlook.com (L.-H.C.)

**Keywords:** multi-omics, trait mining, biological elements, integrative approaches, crop improvement

## Abstract

The advancement of multi-omics tools has revolutionized the study of complex biological systems, providing comprehensive insights into the molecular mechanisms underlying critical traits across various organisms. By integrating data from genomics, transcriptomics, metabolomics, and other omics platforms, researchers can systematically identify and characterize biological elements that contribute to phenotypic traits. This review delves into recent progress in applying multi-omics approaches to elucidate the genetic, epigenetic, and metabolic networks associated with key traits in plants. We emphasize the potential of these integrative strategies to enhance crop improvement, optimize agricultural practices, and promote sustainable environmental management. Furthermore, we explore future prospects in the field, underscoring the importance of cutting-edge technological advancements and the need for interdisciplinary collaboration to address ongoing challenges. By bridging various omics platforms, this review aims to provide a holistic framework for advancing research in plant biology and agriculture.

## 1. Introduction

The rapid rise in global population and the escalating unpredictability of climate patterns have intensified the urgency of improving crop productivity and quality, necessitating more robust and efficient strategies in agricultural science [1]. One of the primary challenges faced by researchers and breeders of staple crops including rice, wheat, maize, and sugarcane is how to improve yield, quality, and survival rates despite the constantly changing conditions brought about by both biotic and abiotic environmental stresses. With the rapid advancements in sequencing and marker technologies, along with the widespread adoption of genome-based breeding methods [2], substantial investments have been made in multi-omics studies on crops. These efforts are bolstered by sophisticated algorithms and powerful computational resources, revolutionizing crop breeding from traditional phenotype-based selection to genomics-assisted breeding and genetic engineering [3,4].

Advances in next-generation sequencing, biomolecular detection technologies, and bioinformatics have catalyzed significant progress across genomics, resequencing, functional genomics, epigenomics, transcriptomics, proteomics, metabolomics, ionomics, and microbiomics, transforming our approach to crop improvement (Figure 1). These omics approaches have become integral to crop improvement efforts, enabling more practical and precise elucidation of underlying genetic mechanisms and their influence on trait development [5]. This transformative research encompasses a broad spectrum of topics, ranging from fundamental plant physiological processes to specific experimental goals aimed at identifying the most sensitive and altered molecular components under varying conditions. The integration of these methods has markedly advanced all stages of the breeding process, from the discovery of new genetic variations to more comprehensive and detailed phenotypic analyses, and the elucidation of important biological elements (including critical genes, transcription factors, and regulatory proteins) related to growth, disease resistance, stress response, and metabolic traits. Extensive multi-omics research has provided valuable insights into intriguing phenotypes and their adaptability to diverse environments. This knowledge is crucial for improving crop varieties by endowing them with adaptive traits. Recent years have witnessed a substantial reduction in the cost of generating multi-omics datasets, enabling the development of extensive interconnected datasets that provide a holistic view of crop biology. These datasets capture the features and impacts of genes, proteins, metabolites, and other components through numerous replicated samples under different experimental conditions. The comprehensive nature of these datasets enables a deeper understanding of critical biological elements and intricate molecular networks, facilitating the development of crops that are more resilient and productive in the face of global challenges.

In this review, we summarize the current developments in various omics technologies and their recent advances in agronomic discovery of biological elements associated with important agronomic traits, and discuss the challenges currently encountered and prospects for the future.

## 2. Principles and Methods of Multi-Omics Technologies

Multi-omics technologies, encompassing genomics, epigenomics, transcriptomics, metabolomics, ionomics, microbiomics, and beyond, provide an expansive and detailed perspective on the multifaceted traits of organisms. In the realm of crop research, these technologies are indispensable for elucidating the genetic underpinnings, environmental adaptability, and developmental processes of crops. By integrating data across various biological layers, multi-omics approaches enable a holistic understanding of crop biology, which is crucial for advancing agricultural science and improving crop performance (Figure 1).

To illustrate the integration of multi-omics approaches, we use rice as an example, highlighting how these technologies can be applied to enhance crop research. First, the assembly and analysis of gap-free reference genome sequences for the elite rice varieties Zhenshan 97 and Minghui 63 have provided a model for studying heterosis and yield in rice [6]. The availability of complete genome sequences significantly improves the quality of genome annotation, enabling researchers to better understand the complexity of genome structures, such as repetitive sequences, structural variations, and chromosomal rearrangements. With the rapid advancement of high-quality pan-genomes, researchers can more accurately identify and analyze large structural variations (SVs) and small single nucleotide polymorphisms (SNPs) that have significant phenotypic effects [7]. By conducting genome-wide association studies of structural variations (SV-GWAS) and single nucleotide polymorphisms (SNP-GWAS), it becomes possible to reveal associations between these variations and complex traits. However, genomics alone may not provide a comprehensive understanding of how genetic variations manifest in biological functions. Therefore, integrating other omics techniques, such as transcriptomics, metabolomics, and epigenomics, can offer a more holistic view of the biological impact of genetic variations, thereby overcoming the limitations of genomics research. For instance, combining transcriptomic and translatomic data from Zhenshan 97 and Minghui 63 enabled the identification of key genes with allele-specific translation efficiency [8]. These genes can be targeted in molecular breeding to enhance the performance of hybrid rice. Another study utilized ATAC-seq to map chromatin accessibility in six tissues of Zhenshan 97, while mGWAS identified regulatory loci for numerous metabolites, linking genotypes with phenotypes and deepening our understanding of gene regulation, thereby supporting trait improvement through breeding [9]. Further research on Zhenshan 97 and Minghui 63 identified lncRNAs and circRNAs acting as competing endogenous RNAs, with the potential to regulate gene expression. For example, osa-156 l-5p (related to yield) and osa-miR444a-3p (related to nitrogen/phosphorus metabolism) were confirmed through dual-luciferase reporter assays to play crucial roles in rice growth and development, laying the foundation for future rice breeding analyses [10]. Although microbiomics alone can characterize microbial communities, integrating it with metabolomics offers a more comprehensive perspective on plant–microbe–environment interactions. This holistic approach enables the development of more effective stress-resistance breeding strategies. In studies of rice pathogens, GWAS identified a novel gene, OsTPS1, which significantly enhances rice resistance to white leaf blight [11]. Finally, combining spatial metabolomics with transcriptomics or proteomics allows researchers to explore the dynamic distribution of metabolites and proteins within plant tissues. This spatiotemporal resolution is essential for understanding the complex regulatory networks that govern plant development and stress responses [12,13]. In summary, while single-omics approaches offer valuable insights, they often fall short of capturing the complexity of biological systems. Multi-omics integration addresses these limitations by providing a more comprehensive understanding of gene function, regulatory mechanisms, and phenotype expression, ultimately enhancing the precision and effectiveness of genetic breeding strategies.

### 2.1. Elucidating Gene Function and Genetic Variation in Crops

Genomics is the scientific study of the structure and function of the genome of an organism, with a focus on structure, function, evolution, mapping, epigenome, mutant genes, and genome editing [14]. With the advent of first-generation sequencing technologies and the development of next-generation sequencing technologies, researchers can now rapidly obtain the complete genome sequences of crops [15]. Genome assembly reconstructs entire genome sequences from short sequencing reads, while genome annotation predicts functional regions within the genome, such as protein-coding genes, regulatory elements, and non-coding RNAs [16]. Recent advances in long-read sequencing technologies have further enhanced the accuracy of these processes, particularly in resolving complex genomic regions [17]. Genomics plays an indispensable role in elucidating genome structure and function, including identifying the locations of key genes and genetic variations, enabling researchers to pinpoint genes associated with specific traits, and providing molecular targets for crop genetic improvement. With the advancement of third-generation sequencing technologies like PacBio single-molecule real-time (SMRT) sequencing and Oxford Nanopore Technologies (ONT) ultra-long sequencing, coupled with the reduction in sequencing costs, an increasing number of telomere-to-telomere genomes and pan-genomes have been published [7,18,19,20,21]. As vast amounts of genomic data are generated, comparative genomic analysis has become exceedingly important. Numerous comparative genomic tools have emerged, aiding in our understanding of candidate genes influenced by variations [22,23,24].

Quantitative trait loci (QTL) mapping and genome-wide association studies (GWAS) are vital for understanding crop traits, with linkage analysis for QTL mapping serving as the direct precursor to association studies. GWAS were first demonstrated as an effective method for identifying genes associated with human diseases [25]. Over the subsequent decades, GWAS have evolved into a powerful and widely used tool for elucidating complex traits, a progression primarily driven by advancements in genomic technologies that enable comprehensive examination of genetic variations across entire genomes within diverse populations [26,27]. Building on this foundation, GWAS have been instrumental in identifying numerous loci linked to key agronomic traits in various crops, such as grain yield in rice and drought resistance in maize [28,29]. These findings underscore the significance of GWAS in crop improvement. Various sophisticated methodologies have been developed to enhance the statistical power and computational efficiency of GWAS, facilitating the detection of genomic variations linked to traditional agronomic phenotypes as well as biochemical and molecular traits [30,31,32]. These associations have significantly advanced gene cloning efforts and enabled the application of marker-assisted selection and genetic engineering, thereby expediting the process of crop breeding and improvement [33]. The rise of high-throughput whole-genome sequencing technologies has transformed these methods into essential tools for cloning and identifying QTL in crops [34]. If alleles are included in existing QTL mapping, QTL has proven to be highly useful for GWAS, serving as a complementary tool in the prioritization process of candidate loci [35]. Moreover, the integration of high-throughput sequencing with advanced genotyping platforms has markedly improved the resolution and accuracy of QTL mapping and GWAS, particularly in identifying rare variants associated with complex traits [36]. QTL mapping associates complex phenotypes with molecular marker data [37], while GWAS identify associations between genomic variations and traits [38]. These methods provide powerful tools for understanding the genetic background and complex traits of crops.

### 2.2. Investigating Epigenetic Regulation and Its Influence on Gene Expression

Epigenetics involves heritable changes other than DNA sequence alterations and focuses on genome-level modifications, mainly histone modifications, DNA methylation, and other techniques [39]. Epigenomics studies the full set of these modifications in the cytogenetic material, revealing how factors associated with the growth of an organism, including environment and stress, affect gene expression and influence crop phenotypes [14]. These epigenetic modifications can lead to changes in plant traits without altering the underlying genetic code. Epigenome maintenance is a continuous process that plays a key role in maintaining the stability of eukaryotic genomes by participating in biological mechanisms such as DNA repair [40]. By studying epigenomics, researchers can better understand crop adaptation to environmental changes and the maintenance of genetic diversity.

Histone modifications have been extensively studied using chromatin immunoprecipitation (ChIP) technology combined with DNA microarrays (ChIP-Chip) [41], providing insights into chromatin dynamics and gene regulation in plant genomes. However, with the advent of CUT&Tag, these analyses can now be performed with less starting material and at higher resolution, offering enhanced precision in genome-wide epigenomic profiling, even from limited plant tissue samples [42]. DNA methylation is a crucial chromatin modification in plant genomes that is mitotically and sometimes meiotically heritable, and it is often associated with gene expression and phenotypic variation [43]. The advent of sodium bisulfite conversion coupled with high-throughput sequencing in 2008, known as whole-genome bisulfite sequencing (WGBS), propelled the field into the single-base resolution era, significantly enhancing our understanding of how genomes employ DNA methylation [44]. In plants, DNA methylation occurs in three distinct contexts: CG, CHG, and CHH, each governed by distinct enzymatic pathways. These methylation contexts not only influence gene expression but also play critical roles in regulating transposable elements and maintaining genomic integrity [45]. Although whole-genome bisulfite sequencing (WGBS) has been available since 2008, it remains the most widely used method for studying 5-methylcytosine today [46]. WGBS have been employed to sequence the methylomes of numerous crop species, including Oryza sativa [47], Zea mays [48], and Brassica napus [49]. This approach has provided significant insights into the role of DNA methylation in crop biology. Chromatin accessibility, a key indicator of the openness of genomic regions to transcription factor binding, is crucial for understanding gene regulation, particularly under varying environmental conditions. Methods such as MNase-seq, DNase-seq, ATAC-seq, and FAIRE-seq are used to analyze the accessible chromatin landscape of cells [50,51,52,53]. These methods are used to map genome-wide epigenetic profiles at single-base resolution by selectively isolating histone-bound or unbound DNA fragments and performing sequencing and reference genome comparisons [54]. Through these advanced technological tools, researchers are able to gain a deeper understanding of the mechanism of epigenetics in the formation of important crop traits, thus providing a theoretical basis and technical support for crop genetic improvement and environmental adaptation.

### 2.3. Characterizing Gene Expression Profiles and Regulatory Networks

Transcriptomics is a technology used to study the sum of all RNA transcripts of an organism, providing direct insight into real-time gene expression profiles. As sequencing technology has advanced, a range of powerful and precise techniques has emerged. Two mainly contemporary technologies are microarrays, which quantify a set of predetermined sequences [55], and RNA-Seq, which uses high-throughput sequencing to record all transcripts [56]. Among these, RNA-Seq has emerged as a powerful and effective method for conducting large-scale transcriptome studies, especially in most non-model plants that lack high-quality reference genomes [57]. Building on these methods, various integrative approaches have emerged. For instance, transcriptome-wide association studies (TWAS) have been utilized to analyze 275 young rice panicle transcriptomes, revealing thousands of genes associated with panicle traits [58]. This approach sheds light on regulatory variations that influence panicle architecture and provides valuable insights into causal genes and gene regulatory networks in rice. Moreover, the development of a rice pan-transcriptome has facilitated the characterization of transcriptional regulatory landscapes in response to cold stress [59]. This highlights the complexity of transcriptomic responses and underscores the importance of pan-transcriptomes in capturing the full spectrum of genetic diversity and regulatory mechanisms under stress conditions.

Processing transcriptome data, particularly from RNA-Seq, demand significant computational resources due to the vast amount of parallel sequencing reads generated, necessitating advanced bioinformatics tools for accurate analysis [60]. When studying a particular species, transcriptomic datasets are often used for gene co-expression analysis (e.g., weighted gene co-expression network analysis (WGCNA)) by methods such as Spearman correlation coefficient (SCC), Pearson correlation coefficient (PCC), and mutual rank (MR) [61]. In order to identify unknown biosynthetic genes in a target pathway, key decoy genes are usually required [62]. Successful co-expression analysis hinges on the accurate correlation between regulatory and functional genes, which is essential for elucidating gene networks involved in key biological processes. The reliability of transcriptome data can be validated through quantitative PCR (qPCR) [63]. Functional validation is typically achieved through gene knockout or rescue experiments [64]. These analytical steps ensure the accuracy and biological relevance of transcriptomic data, which helps to discover genes of interest and reveal mechanisms of gene expression regulation.

### 2.4. Deciphering Protein Interaction Networks and Functional Proteomes

Metabolomics is considered the phenotypic endpoint of histological studies and aims to capture the end result of information transfer from the genome to the transcriptome and proteome through comprehensive qualitative and quantitative analyses of all small molecules in an organism [65]. It provides a snapshot of the metabolic state of an organism, reflecting the biochemical activities and physiological status at a given time. Studies in this field cover the chemical processes of metabolites, small molecule substrates, intermediates, and cellular metabolites, aiming to reveal changes in metabolic pathways that may affect specific traits by analyzing metabolic small molecules in organisms [66,67].

Metabolite determination methods in plant research are primarily categorized into targeted and non-targeted approaches. Targeted analysis focuses on quantifying specific metabolites by comparing them to known standards. This approach is highly sensitive and specific, making it suitable for hypothesis-driven studies where the metabolites of interest are predefined. In contrast, non-targeted analysis aims to discover and identify as many metabolites as possible without prior knowledge of their identity. This approach compares metabolites based on their relative intensities, providing a comprehensive overview of the metabolome and enabling the discovery of novel metabolites and pathways. Currently, metabolomics techniques rely on ultra-high-pressure liquid chromatography (UHPLC) combined with high-resolution mass spectrometry (HRMS) or NMR spectroscopy techniques [68]. These advanced techniques are capable of providing detailed chemical information for accurate analysis and characterization of thousands of compounds [65]. These techniques are further divided based on the detection methods used. Gas chromatography–mass spectrometry (GC-MS) is primarily employed for detecting volatile compounds, while liquid chromatography–mass spectrometry (LC-MS) is used for compounds that are less volatile and exhibit poor thermal stability [69,70]. For instance, integrating genomic imprinting and metabolomic analyses in rice, such as those identifying flavonoid accumulation and stress tolerance genes [71,72], has unveiled pathways and genetic regulators that can be leveraged to boost disease resistance, seed vigor, and nutritional content [73,74,75]. These insights provide a valuable foundation for future research aimed at enhancing plant traits through metabolomic studies.

### 2.5. Analyzing Plant-Microbiome Interactions and Microbial Diversity

Plant microbiomics, also known as phytomicrobiomics, has rapidly emerged as a burgeoning field of research in recent years, owing to microorganisms playing a crucial role in plant health and productivity [76]. The plant microbiome not only forms the basic foundation for plant growth but also plays a key role in enhancing plant resistance to environmental stresses and diseases. It has been shown that plants form complex symbiotic relationships with a variety of microbial communities that play important roles in plant physiology and ecology [76,77]. These microbial communities inhabit the interior (endophytes) or surface (epiphytes) of plant tissues [78]. In particular, inter-root microbial communities are located at the interface between roots and soil, facilitating the uptake of mineral nutrients by plants and helping to defend against pathogen invasion. For instance, nitrogen and phosphorus uptake in legumes is largely facilitated by symbiotic relationships with arbuscular mycorrhizal fungi, which enhance nutrient acquisition from the soil [79].

Recent advancements in high-throughput sequencing and metagenomic techniques have significantly propelled our understanding of plant microbiomes, particularly in uncovering microbial contributions to plant health and productivity under diverse environmental conditions [80]. Through metagenomics, researchers have been able to delve into the composition, function, and dynamics of the plant microbiome under different environmental conditions. Such studies have not only elucidated the complex interaction mechanisms between plants and microbes but also provided new perspectives on agricultural practices. A key discovery in plant microbiome research is the role of specific microorganisms in bolstering plant resistance to biotic and abiotic stresses, paving the way for the development of microbial-based biofertilizers and biopesticides [81]. For example, research on plant microbiomes has revealed the enhancement of plant resistance by specific microorganisms, which provides a theoretical basis for the development of new biofertilizers and biopesticides [82,83,84]. In addition, by understanding how microbes affect plant growth and development, researchers can design more sustainable agricultural management strategies.

## 3. Exploring Biological Elements in Crop Research Through Integrative Multi-Omics Approaches

Multi-omics technologies are central to advancing modern crop research, employing diverse strategies to pinpoint biological elements crucial to trait development and adaptation (Figure 2). Key biological components are identified and characterized through a thorough exploration of crop traits, encompassing agronomic performance (yield, quality, flavor, texture, etc.), responses to abiotic stresses (temperature extremes, drought, flooding, salinity, high light intensity, heavy metal toxicity), and resilience against biotic threats (fungi, bacteria, viruses, parasites, insects, weeds). These integrated investigations are pivotal for transformative innovations in agriculture. Multiple multi-omics analytical methods are listed in Table 1. This integrated approach not only promotes the effective utilization of crop genetic resources but also accelerates the innovation and implementation of novel crop breeding strategies, bringing an important driving force for global food security and sustainable agricultural development.

### 3.1. Exploration of Agronomic Traits

Grain yield is a primary concern for many researchers. Multi-omics approaches have identified numerous QTLs related to yield in rice, wheat, and maize [141,142,143,144,145]. In rice, comprehensive studies have identified critical QTL influencing traits like grain length and weight, notably with loci such as qGL11 associated with OsGH3.13 [146]. In wheat, the integration of QTL mapping with WGCNA has uncovered candidate genes influencing plant height and spike length, while epigenomic approaches have clarified regulatory elements that modulate yield-related traits [147,148]. In maize, QTLs like qKRN2 have been linked to ear row number, with priority genes identified through the integration of interlocking populations and transcriptomic data [149]. In other crops, multi-omics approaches have also uncovered significant genetic insights. For example, transcriptomic data revealed that taller coconut varieties express more lignin biosynthesis genes, such as CCR and F5H, with GWAS confirming key SNPs in the promoter region of the GA20ox gene on chromosome 12 as regulators of height variation [150]. In sea island cotton, bulked segregant analysis sequencing (BSA-seq), RNA-seq, and whole-genome resequencing analyses identified the qD07-NB locus on chromosome D07, linking a missense SNP in the candidate gene Gbar_D07G011870 to the nulliplex-branch trait [151]. Thus, while QTL mapping identifies loci associated with crop yield, the approach yields numerous loci with varying precision. The integration of genomic, transcriptomic, and epigenomic approaches enables the precise identification of key genes affecting crop yield, thereby reducing both breeding time and costs.

Oil content is a critical trait in oilseed crops like oilseed rape, peanuts, and soybeans. Researchers have utilized various technologies to advance understanding in this area. For instance, a high-density genetic map of peanuts was constructed using simplified genome sequencing of 120 samples, identifying 27 QTLs associated with kernel weight and size [152]. Integrated approaches, including QTL-seq, QTL mapping, and RNA-seq, subsequently pinpointed major QTLs related to peanut seed weight [153]. In Brassica napus, GWAS, TWAS, genomic selection, and gene module analysis on 505 inbred lines identified QTLs, genes, and regulatory networks associated with seed oil content (SOC) [154]. Additionally, integrative analyses of transcriptomics, proteomics, and metabolomics identified spermidine synthase in soybean seeds, offering insights for enhancing seed oil content through molecular breeding strategies [155].

Besides yield, the flavor, texture, color, and nutritional content of a crop are key traits that contribute to its excellence. In tomatoes, studies utilizing advanced algorithms and integrating transcriptomic, lipidomic, and metabolomic data have clarified regulatory mechanisms, such as AtMYB12’s involvement in flavonoid synthesis [156] and SlERF.H6’s role in reducing bitterness [157]. Similarly, integrative approaches in coconut [158], citrus [159], passion fruit [160], kale [161], cashew [162], and green pepper [163] have identified pivotal genes and regulatory networks influencing traits such as lipid synthesis, flavonoid production, aroma compounds, and nutrient metabolism. These studies underscore the powerful potential of multi-omics technologies in crop physiology and quality control, revealing complex metabolic regulatory networks and gene expression mechanisms, which provide valuable scientific insights and methodological tools for improving crop quality.

### 3.2. Understanding Adaptation to Various Environmental Conditions

Plants have evolved complex mechanisms to adapt to environmental stresses such as drought, extreme temperatures, high salinity, and heavy metal exposure. They detect stress signals through receptors and initiate signaling pathways. These pathways activate stress response genes through secondary messengers, resulting in the development of specific stress adaptation mechanisms at the transcriptional and translational levels. In studies on plant drought and heat tolerance, researchers have focused on identifying key physiological and molecular mechanisms that enable plants to withstand prolonged periods of water scarcity. In rapeseed, integrating GWAS and RNA-seq analysis of 119 varieties revealed novel SNPs linked to the drought tolerance gene ABCG16 [164]. Similarly, analyses of Nagina 22 rice highlighted the role of auxiliary carbohydrate metabolism and L-phenylalanine biosynthesis in drought tolerance [165]. In tomatoes, a metabolome genome-wide association study (mGWAS) identified gene clusters regulated by SlMYB13, enhancing drought resistance through phenylpropanoid metabolism [166]. Maize studies revealed ZmHB77 as crucial for drought tolerance by regulating root architecture [167]. Additionally, comprehensive multi-omics approaches in maize have identified numerous QTLs and significant SVs associated with drought tolerance [168].

Recent advancements in plant cold tolerance research have significantly progressed through detailed studies on physiological responses and genetic mechanisms, aiming to enhance resilience in adverse climates. In rice, GWAS identified a QTL linked to seedling cold resistance, with the OsSEH1 gene playing a pivotal regulatory role [169]. Comprehensive transcriptomic and metabolomic analyses demonstrated that OsSEH1 orchestrates gene expression and metabolite accumulation in the phenylpropanoid and flavonoid biosynthesis pathways. Additionally, the heightened sensitivity to exogenous abscisic acid (ABA) observed in the osseh1 mutant suggests that OsSEH1 regulates cold hardiness through ABA signaling pathways [170]. In wheat, proteomic studies focusing on acetylation, complemented by multi-omics analyses, have identified the wheat cold stress-responsive protein TaPGK, underscoring its positive regulatory function in cold tolerance [171]. Regarding peanuts, BSA-seq has been employed to identify QTLs and genes associated with cold tolerance during the seedling emergence stage [172].

Understanding the strategies plants use to manage high salinity is crucial for developing salt-tolerant crops and enhancing agricultural sustainability. In Arabidopsis thaliana, mutants have demonstrated enhanced salt tolerance through the accumulation of stress-related metabolites [173]. GWAS in wheat have identified loci associated with salt tolerance traits [174]. Comprehensive metabolomics, proteomics, hormone, and ion analyses on date palms have highlighted mechanisms of salt avoidance and adaptation, including ion excretion and osmotic regulation [175]. These findings provide essential molecular markers and genetic insights for crop breeding aimed at enhancing salt tolerance. Nitrogen uptake remains pivotal for crop productivity. In rice, GWAS have linked OsGATA8 to nitrogen uptake efficiency, influencing tillering [176]. Research in *Brassica rapa* utilized multi-omics approaches to identify interactions within agricultural ecosystems, emphasizing the role of soil organic nitrogen in crop yield [177]. In maize, extensive integration of multi-omics data has been employed to predict genes associated with nitrogen-use efficiency [178].

These extensive studies unveil the complex genetic, molecular, and physiological adaptations that enable plants to survive environmental stresses. Through the integration of multi-omics approaches and cutting-edge genomic techniques, they advance our understanding of plant resilience and guide the development of robust crop varieties crucial for sustainable agriculture in the face of climate change.

### 3.3. Enhancing Resistance to Biological Stresses

Plants have developed complex defense mechanisms to counter biotic stresses such as fungi, bacteria, viruses, parasites, and insects. They detect stress signals via specific receptors, activating pathways that trigger stress response genes. These genes are then expressed to mount effective defenses against various biotic stresses. Microbial interactions with plants are pervasive, driving extensive research aimed at deciphering these dynamics. In rice blast disease research, integrating multi-omics data with WGCNA and graph autoencoder techniques has revealed crucial Magnaporthe oryzae Oryzae small RNAs, rice genes, mRNAs, and proteins involved in significant biological processes [179]. In another study, populations of chromosome segment substitution lines (CSSLs) in rice were used to identify QTLs and structural variants associated with rice blast resistance. This included the gene LOC_Os07g35680, which exhibits increased expression due to a 7.8 kb insertion in its wild allele [180]. In studies of other rice pathogens, GWAS of over 200 rice populations identified a novel gene, OsTPS1, involved in synthesizing the sesquiterpene α-erythromycetin. It was demonstrated that OsTPS1 is epigenetically regulated by JMJ705 via the methyl jasmonate pathway, significantly enhancing rice resistance to white leaf blight [11]. Similarly, studies in cotton revealed the MYB transcription factor RVE2 as key to Verticillium dahliae resistance [181]. Research in Fagopyrum tataricum highlighted genes linked to resistance against Rhizoctonia solani, emphasizing the role of cytochrome P450 in flavonoid accumulation [182].

Insects and pests pose significant challenges to crop cultivation. Early studies utilizing comparative proteomic and transcriptomic analyses identified 352 genes encoding secreted proteins from the salivary glands of rice pests TN1 and Mudgo, which interact with rice to influence its growth [183]. Key proteins, including endo-β-1,4-glucanase (NlEG1) and NlSEF1, have been identified in the interaction between rice and the brown planthopper, underscoring their role in rice defense mechanisms [184,185]. Shi et al. combined transcriptomics and metabolomics to compare Bph30 transgenic rice (BPH30T) with susceptible Nipponbare rice under brown planthopper (BPH) infestation, revealing that Bph30 likely enhances resistance by facilitating metabolite and hormone transport via the shikimic acid pathway [186]. Additionally, the yellow stem borer (YSB) is a major threat to rice. Gokulan et al. used bulk-segregant analysis and next-generation sequencing to map a QTL interval for YSB resistance in the rice line SM92. Their transcriptome and metabolome analyses suggested a link between phenylpropanoid metabolism and YSB resistance, providing insights into plant defense mechanisms against this pest [187].

Recent advancements in multi-omics technologies have profoundly deepened our understanding of plant–pathogen interactions, uncovering crucial molecular mechanisms of host resistance and pathogen virulence. These integrated approaches offer promising strategies for enhancing plant health and promoting agricultural sustainability.

## 4. Emerging Technologies, Challenges, and Future Prospects

In recent years, the rapid advancement of multi-omics technologies has significantly transformed the landscape of biological research, with single-cell omics and spatial omics emerging as particularly groundbreaking fields. These advancements provide unprecedented opportunities to investigate cellular and molecular processes at an unparalleled resolution, enabling more precise dissection of biological mechanisms. Nevertheless, they also pose substantial challenges, including the requirement for advanced data integration techniques, substantial computational resources, and the innovation of novel experimental methodologies (Figure 3) [188,189]. Addressing these challenges will be pivotal for fully harnessing the potential of these technologies, thereby facilitating groundbreaking discoveries and pioneering applications across disciplines such as medicine, agriculture, and environmental science.

### 4.1. Advances in Single-Cell and Spatial Omics

The concept and technology of single-cell RNA sequencing (scRNA-seq) were first introduced in 2009 [190]. Single-cell multimodal omics and spatially resolved transcriptomics technologies were named Method of the Year by *Nature Methods* in 2019 and 2020, respectively [191,192]. This pioneering approach allowed researchers to explore gene expression at the single-cell level, revealing cellular heterogeneity within tissues that was previously undetectable with bulk RNA sequencing methods.

Following this initial breakthrough, a series of innovative sequencing technologies combining single-cell approaches with other omics disciplines have emerged. These advanced methods, including single-cell multi-omics and spatial transcriptomics, represent cutting-edge techniques for dissecting the complex architecture of tissues, organs, and entire organisms. They focus on identifying distinct cell types, characterizing their specific functions, and understanding how cellular interactions shape biological systems [193]. While these methods have greatly improved the resolution and accuracy of scRNA-seq, the core principle of studying individual cells to uncover cellular heterogeneity remains a cornerstone of this field. As cells are considered the structural units of life, understanding the differences between cell types and their developmental trajectories is crucial for gaining deeper insights into the fundamental processes of life.

One of the primary challenges with scRNA-seq is the requirement for tissue dissociation, which inevitably leads to the loss of spatial information. Spatial information is essential for understanding the interactions and regulatory processes between cells during development and their interactions with the environment. This has driven the development of spatial multi-omics, which aims to retain and utilize spatial context in conjunction with molecular data [194]. Spatial transcriptomics (ST), for instance, maps the spatial distribution of cells within tissues and reveals local communication networks, underscoring the importance of integrating spatial and single-cell data to achieve a more comprehensive understanding of cellular dynamics and tissue organization [195].

### 4.2. Applications of Plant scRNA-Seq and ST

Single-cell RNA expression profiling has rapidly become an indispensable method in various research fields involving humans, animals, and plants. This technology allows for unprecedented accuracy and speed in identifying rare and novel cell types within tissues, offering significant advantages over traditional bulk RNA sequencing methods [196]. Due to these characteristics, scRNA-seq has been effectively used in disease diagnosis, therapeutic strategy development, and the exploration of developmental biology [197].

In plant research, the gene expression patterns governing cellular development often vary considerably across distinct developmental stages, emphasizing the importance of single-cell analysis for uncovering these temporal changes [198]. Studying these processes at the single-cell level is thus critical for unraveling the intricate mechanisms that drive plant development and differentiation. Several research groups have utilized high-throughput scRNA-seq and ST to study Arabidopsis thaliana, the most widely used model plant in molecular genetics [199,200,201,202]. Other model plants, such as rice [12,13], tomato [203], and maize [204,205], have also been the subjects of extensive single-cell and ST studies.

The availability of various web-based graphical resources for plant scRNA-seq data has further facilitated the accessibility and usability of these data for researchers. For example, detailed graphical information on plant scRNA-seq data can be accessed online, providing valuable insights and tools for further research [206]. Despite significant progress in scRNA-seq, the inherent loss of spatial context continues to be a critical limitation. Early approaches, including in situ hybridization (ISH), single-molecule RNA fluorescence ISH, and laser capture microdissection (LCM), were employed to mitigate the spatial information loss inherent to scRNA-seq [207,208,209]. However, these methods have not been widely adopted in plant research due to challenges such as the difficulty of plant cell wall hydrolysis and the diffusion of intracellular transcripts to the array surface.

In 2017, Giacomello et al. made significant advancements by optimizing tissue fixation, staining, and permeabilization steps in ST technology [210]. This led to the successful creation of the first spatial gene expression maps of whole transcriptomes in Arabidopsis thaliana inflorescence meristems, developing and dormant leaf buds of Populus tremula, and female cone buds of Abies fabri, demonstrating the feasibility of this technology in plant research. Following this, Ståhl et al. established a comprehensive spatial expression map of Arabidopsis, and Xia et al. characterized Arabidopsis leaves using single-cell Stereo-seq [211,212]. Moreover, Stereo-seq has been used to construct spatiotemporal maps of other model organisms, such as Mus musculus, Drosophila, and Brachydanio rerio, highlighting its versatility and potential for broad applications [213,214,215].

### 4.3. Challenges and Future Prospects

The integration of multi-omics data presents significant challenges due to the heterogeneous nature of the data across different platforms. One of the key challenges is the alignment and normalization of data originating from diverse sources, each with its own inherent biases, noise, and scaling issues. Additionally, the complexity of biological networks and the dynamic interplay between omics layers complicate the interpretation of integrated results, requiring advanced computational models that can handle large-scale, high-dimensional data while maintaining biological relevance. The early discoveries and applications of scRNA-seq were predominantly achieved in animal and human cells, leaving many challenges to be overcome in plant research (Figure 3) [216]. However, lessons learned from animal and human studies can help pave the way for advancements in plant research. Analyzing the application of these technologies in animal experiments can provide valuable guidance for designing experiments in plants.

Single-cell gene expression data frequently exhibit high levels of noise, leading to erroneous clustering where cells of the same type may separate, and cells of different types may cluster together due to batch effects [217]. This considerable noise poses significant challenges for data analysis and interpretation. Despite extensive research, several challenges remain for computational data integration, necessitating the development of new and improved integration methods [218]. Furthermore, transcription is not the sole factor influencing plant development. Protein and metabolite levels also play crucial roles in regulating developmental processes. Therefore, a comprehensive approach that includes studying various cellular components is essential for accurately identifying the key factors involved in plant development. Spatial multi-omics technologies, which integrate metabolic and genetic information during development, enable the analysis of correlations between key metabolites and gene expression at single-cell resolution. Through spatial multi-omics, researchers can observe cell–cell interactions, gain deeper insights into cellular functions, identify rare cell populations, and characterize complex metabolites, thereby laying a strong foundation for advancing plant developmental biology [219].

scRNA-seq has been proven to be one of the most transformative technologies in the life sciences, with applications spanning almost all areas of biological research. As the technology continues to evolve, significant progress is expected in several key areas, including increased throughput, reduced costs, and the incorporation of more modalities in a single assay. A promising future direction for scRNA-seq lies in its integration into routine clinical diagnostics and personalized medicine, where it could revolutionize disease classification and treatment strategies. Traditionally, the classification of cells and tissues has been based on their structure and function. To better understand the evolutionary and developmental relationships between tissues or cell types across species, conducting single-cell transcriptomic analyses across different species is essential [220]. Two primary challenges in single-cell epigenomics analysis are the dissociation of cells or nuclei, which results in the loss of tissue context information, and the inefficiency and incompleteness of current technologies. Current single-cell epigenomic approaches typically capture only a small fraction of the epigenome per cell and input population. Improving the efficiency and comprehensiveness of these methods is vital for profiling scarce or rare clinical samples, ensuring that meaningful data can be extracted even from limited material [221]. Similarly, two primary challenges in significantly advancing single-cell proteomics can be broadly categorized into the efficient transfer of proteins from individual cells to the MS detector and the enhancement of throughput without compromising coverage comprehensiveness [222]. Despite these challenges, innovations in single-cell technologies are advancing genomics, transcriptomics, epigenomics, and proteomics, providing more profound insights into cellular diversity and functionality across biological systems.

The advent of single-cell multi-omics is expected to revolutionize our understanding of cellular biology by enabling the simultaneous analysis of multiple omics data (genomics, transcriptomics, proteomics, metabolomics, etc.) from the same single cell. This comprehensive approach will provide deeper insights into how cellular-level variations influence ultimate phenotypic traits. Joint analysis of single-cell and other multi-omics data holds the promise of advancing our understanding of complex biological processes, paving the way for new discoveries and innovations in the field of life sciences.

## 5. Conclusions

The integration of multi-omics technologies has profoundly impacted crop research, yielding unparalleled insights into the genetic, molecular, and metabolic underpinnings of key agronomic traits and responses to environmental challenges. Genomics and transcriptomics have enabled precise identification of genes and pathways crucial for yield enhancement and stress tolerance, while proteomics and metabolomics have provided a deeper understanding of metabolic networks and defense responses against biotic and abiotic stresses. These advancements are accelerating breeding efforts and paving the way for more resilient and sustainable agricultural practices in the face of global challenges such as climate change. As we look ahead, overcoming challenges in data integration and computational analysis will be critical to fully harnessing the potential of multi-omics for predicting and manipulating complex traits in crops. By continuing to innovate and collaborate across disciplines, we can ensure a productive, resilient, and sustainable agricultural future, addressing food security needs and environmental sustainability on a global scale.

## Figures and Tables

**Figure 1 ijms-26-01466-f001:**
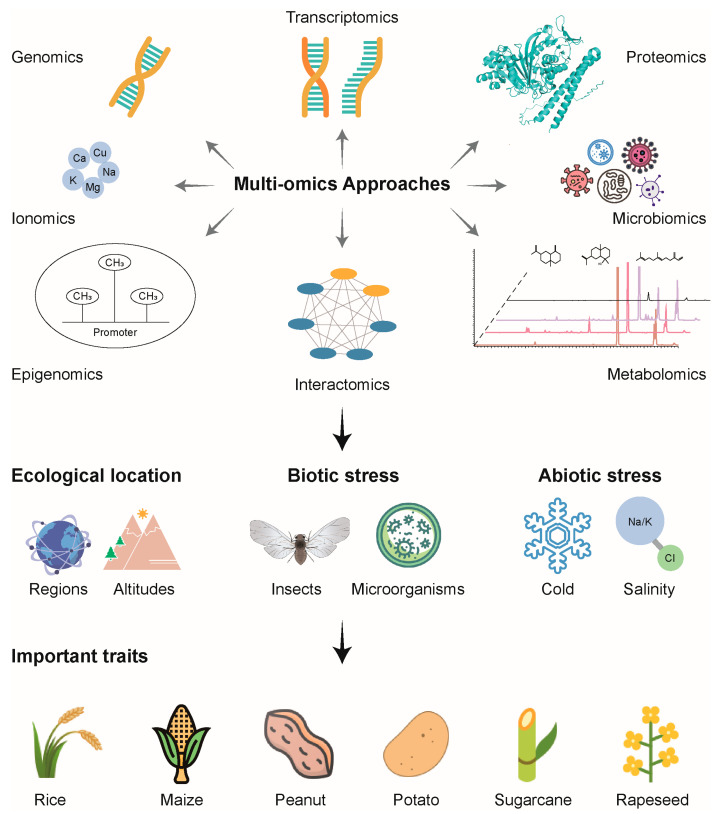
Schematic diagram of the integrative analysis of important traits using multi-omics methodologies.

**Figure 2 ijms-26-01466-f002:**
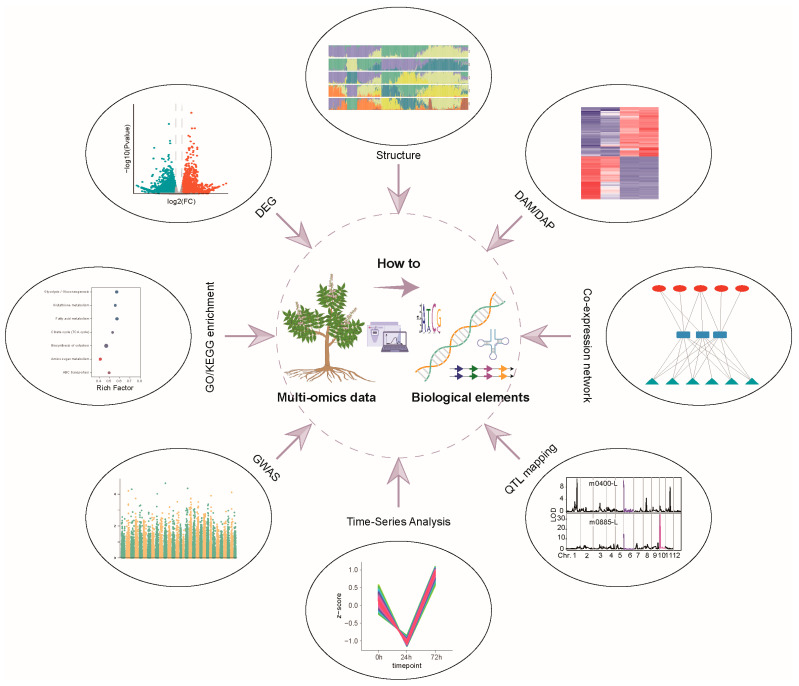
Schematic diagram of various approaches from multi-omics data to biological elements related to important traits. DEG, differentially expressed genes; DAM, differentially accumulated metabolites; DAP, differentially accumulated proteins; GWAS, genome-wide association study.

**Figure 3 ijms-26-01466-f003:**
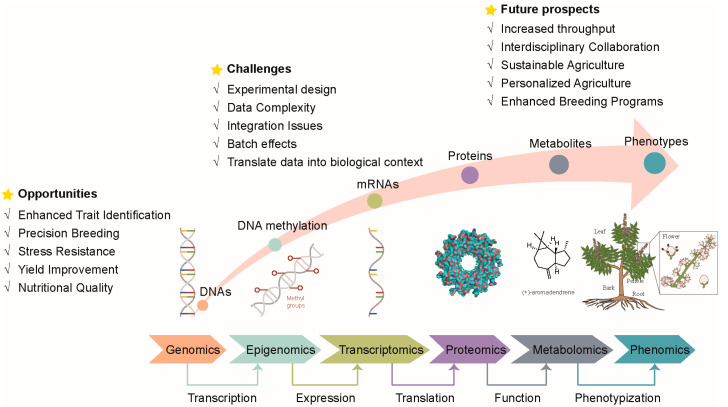
The opportunities, challenges, and future prospects for crop improvement through multi-omics techniques.

**Table 1 ijms-26-01466-t001:** Multi-omics data analysis methods.

Multi-Omics	Method	Reference
Genomics	Sanger sequencing	[85]
	Whole-genome sequencing (WGS)	[86]
	Whole-exome sequencing (WES)	[87]
	Replication sequencing (Repli-seq)	[88]
	PacBio Single-molecule real-time sequencing (SMRT) technology	[20]
	Nanopore DNA sequencing	[21]
Epigenomics	Chromatin immunoprecipitation (ChIP-seq)	[89]
	ChIP-exo	[90]
	Assay for transposase-accessible chromatin (ATAC-seq)	[53]
	Hi-C	[91]
	Chromatin interaction analysis by paired-end tag sequencing (ChIA-PET)	[92]
	Chromatin isolation by RNA purification sequencing (ChIRP-seq)	[93]
	Reduced representation bisulfite sequencing (RRBS-seq)	[94]
	Bisulfite sequencing (BS-seq)	[95]
	Methyl-CpG-Binding Domain Sequencing (MBD-seq)	[96]
	DNAse-seq	[97]
	Parallel Analysis of RNA Structure (PARS)	[98]
	Structure-seq	[99]
	Parallel analysis of RNA ends sequencing (PARE-seq)	[100]
	Massively parallel functional dissection sequencing (MPFD)	[101]
	Methylated RNA immunoprecipitation sequencing (MeRIP-seq)	[102]
	Single-molecule real-time sequencing (SMRT-seq)	[103]
Transcriptomics	RNA sequencing (RNA-seq)	[104]
	Isoform sequencing (Iso-seq)	[105]
	Targeted RNA sequencing	[106]
	Ribosome profiling (Ribo-seq)	[107]
	Global run-on sequencing (GRO-seq)	[108]
	Nascent-seq	[109]
	Native elongating transcript sequencing (NET-seq)	[110]
	PolyA-sequencing (PolyA-seq)	[111]
Proteomics	Mass Spectrometry	[112]
	Liquid Chromatography–Mass Spectrometry (LC-MS)	[113]
	Reverse Phase Protein Array (RPPA)	[114]
	Gel electrophoresis	[115]
	Isobaric tag for relative and absolute quantitation (iTRAQ)	[116]
	Stable isotope labeling by amino acids in cell culture (SILAC)	[117]
Metabolomics	Mass Spectrometry	[118]
	Nuclear Magnetic Resonance (NMR)	[68]
	Liquid Chromatography–Mass Spectrometry (LC-MS)	[70]
	Gas Chromatography–Mass Spectrometry (GC-MS)	[69]
Interactomics	RNA on a massively parallel array (RNAMaP)	[119]
	RNA immunoprecipitation sequencing (RIP-seq)	[120]
	ChIP-Seq	[89]
	Yeast Two-Hybrid (Y2H)	[121]
	Bimolecular fluorescence complementation (BiFC)	[122]
	Crosslinking-immunoprecipitation sequencing (CLIP)	[123]
Ionomics	Inductively coupled plasma mass spectrometry (ICP-MS)	[124]
	Synchrotron X-ray fluorescence (SXRF)	[125]
	Deletion mapping	[126]
	DNA microarray-based bulk segregant analysis (BSA)	[127]
Microbiomics	16S rRNA	[128]
	18S rRNA	[129]
	Internal transcribed spacer (ITS)	[130]
	Shotgun	[131]
	Metagenomics	[132]
Single-cell	Single-cell RNA sequencing (scRNA-seq)	[133]
	scATAC-seq	[134]
	scDNase-seq	[135]
	Single-cell genome-wide bisulfite sequencing	[136]
	Single-cell ChIP-seq	[137]
	Single-cell Hi-C	[138]
	proximity ligation assay for RNA (PLAYR)	[139]
	Mass Spectrometry for single-cell	[140]

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
