# Peer review of "Integrative Multi-Omics Approaches for Identifying and Characterizing Biological Elements in Crop Traits: Current Progress and Future Prospects"

_ijms, 2025, doi:10.3390/ijms26041466_

Round 1

Reviewer 1 Report

Comments and Suggestions for Authors

The article entitled "Integrative multi-omics approaches for identifying and characterizing biological elements in crop traits: Current progress and future prospects" presents a review of integrated multi-omics approaches used in crop research, focusing on the identification of key biological elements associated with crop traits. The authors describe the advances in genomics, transcriptomics, proteomics, metabolomics, and epigenomics, illustrating how the integration of these fields has revolutionized plant breeding and genetic improvement. Particular emphasis is placed on improving yield, stress tolerance and quality by combining omics data using advanced computational methods. The article is suitable for publication in International Journal of Molecular Sciences, I have no major objections to the article; the authors could possibly add a some more information on challenges associated with data integration. A more detailed discussion of the specific computational tools and algorithms used in multi-omics integration would increase the practical value of the paper.

Author Response

Comment 1: The reviewer suggests adding more information on the challenges associated with data integration in multi-omics research.

Response1: We appreciate the reviewer’s suggestion and agree that discussing the challenges in data integration would enhance the comprehensiveness of the article. To address this, we have added a discussion on the challenges of multi-omics integration in Section 4.3, specifically in lines 542-548.

The revised text is as follows: The integration of multi-omics data presents significant challenges due to the heterogeneous nature of the data across different platforms. One of the key challenges is the alignment and normalization of data originating from diverse sources, each with its own inherent biases, noise, and scaling issues. Additionally, the complexity of biological networks and the dynamic interplay between omics layers complicate the interpretation of integrated results, requiring advanced computational models that can handle large-scale, high-dimensional data while maintaining biological relevance.

Comment 2: The reviewer recommends a more detailed discussion of the specific computational tools and algorithms used in multi-omics integration to increase the practical value of the paper.

Response2: We thank the reviewer for this valuable suggestion. In response, we have provided a detailed list of common methods used in multi-omics data analysis in Table 1. These methods are not exclusive to individual omics fields; for example, ChIP-Seq can be applied across epigenomics, interactomics, single-cell, and other omics domains. In the relevant sections, we also mention several algorithms and tools for multi-omics integration, such as WGCNA and PCC, as discussed in Section 2.3. Due to space limitations, we did not further elaborate on these tools, but we believe that referencing the relevant literature will help readers easily find the corresponding algorithms and methods.

Reviewer 2 Report

Comments and Suggestions for Authors

The manuscript provides a comprehensive overview of the current progress and prospects of integrated multifunctional methods for identifying and characterising biological elements of crop traits. The work is interesting and valuable.

Line 117: Incorrect citation (Zhang et al. 2021). Missing references 12 and 13.

Lines 409-410: Brassica rapa should be written in italics.

The reference list does not meet the requirements of the journal Int. J. Mol. Sci. Please arrange it accordingly.

Author Response

We sincerely thank Reviewer 2 for their positive feedback and valuable suggestions, which have greatly helped improve the quality of our manuscript. Below are our responses to the specific points raised:

Comment 1: The reviewer points out an incorrect citation (Zhang et al. 2021) and mentions that references 12 and 13 are missing.

Response1: We appreciate the reviewer’s attention to detail. We have corrected the citation for Zhang et al. (2021) and ensured that references 12 and 13 are now properly included in the manuscript. We have also reviewed and updated the reference list to ensure all citations are accurate and complete.

Comment 2: The reviewer notes that Brassica rapa should be written in italics in lines 409-410.

Response2: We thank the reviewer for this observation. We have corrected the formatting of Brassica rapa to italics in the relevant section (lines 409-410).

Comment 3: The reviewer highlights that the reference list does not meet the requirements of the journal Int. J. Mol. Sci..

Response3: We are grateful for the reviewer’s guidance on this matter. We have carefully revised the reference list to conform to the formatting requirements of the International Journal of Molecular Sciences journal. All references have been adjusted according to the journal's style guidelines.

Reviewer 3 Report

Comments and Suggestions for Authors

Dear Authors,

I am pleased to provide my review of the manuscript entitled "Integrative Multi-Omics Approaches for Identifying and Characterizing Biological Elements in Crop Traits: Current Progress and Future Prospects."

Your article addresses a critical and timely topic. It serves as a much-needed review of multi-omics tools that have transformed methodologies for analyzing complex biological systems in recent years. The comprehensive information you provide on the molecular mechanisms underlying key traits of various organisms is particularly valuable.

You have accurately highlighted that integrating data from genomics, transcriptomics, metabolomics, and other omics platforms enables researchers to systematically identify and characterize genetic elements relevant to plant phenotypes. The insights offered in your review on recent advances in using multi-omics approaches to elucidate genetic, epigenetic, and metabolic networks involved in key plant traits are both relevant and highly informative.

For researchers like myself and others in the field, this article serves as an essential resource, showcasing current trends and opportunities to integrate analytical and computational methods for better planning and execution of research. In an era of rapid scientific advancements, it is often challenging to stay abreast of and implement emerging innovations in laboratory practices. Your review provides a critical evaluation of these developments and offers a holistic perspective that is highly beneficial to the scientific community.

Furthermore, your discussion on the potential of integrative strategies to enhance crop improvement, optimize agricultural practices, and promote sustainable environmental management is insightful. The emphasis on cutting-edge technological advancements and the necessity of interdisciplinary collaboration to address ongoing challenges underscores the relevance of your work.

In conclusion, I believe the publication of this manuscript is well-justified. After addressing linguistic and formal corrections, the article will undoubtedly make a strong contribution to the field.

Best regards,

reviewer

Author Response

We would like to express our sincere gratitude to Reviewer 3 for their thoughtful and encouraging comments. We are pleased to know that you find our article valuable and relevant to the field. Below are our responses to the points raised:

Comment 1: The reviewer has praised the overall quality of the manuscript, particularly the comprehensive information provided on multi-omics tools and their application in understanding plant traits. The reviewer also highlights the manuscript's relevance to ongoing research and its contribution to the scientific community.

Response1: We deeply appreciate your kind words and positive feedback regarding the importance and relevance of our review. We are pleased that the manuscript serves as a valuable resource for researchers in the field. Your comments confirm the importance of integrating multi-omics approaches in plant research, and we are glad that our work effectively conveys these developments.

Comment 2: The reviewer recommends addressing linguistic and formal corrections.

Response2: We thank the reviewer for pointing out the need for linguistic and formal revisions. We have carefully reviewed the manuscript and made the necessary changes to improve clarity, grammar, and style. All minor linguistic and formatting errors have been corrected, and we believe these revisions enhance the overall readability of the article.

We would like to once again express our sincere gratitude to all the reviewers for their invaluable feedback and constructive suggestions, which have greatly contributed to improving the quality of our manuscript.